# Optimization of the Electrospray Process to Produce Lignin Nanoparticles for PLA-Based Food Packaging

**DOI:** 10.3390/polym15132973

**Published:** 2023-07-07

**Authors:** Rodrigue Daassi, Kalvin Durand, Denis Rodrigue, Tatjana Stevanovic

**Affiliations:** 1Renewable Materials Research Centre (CRMR), Institute of Nutrition and Functional Foods (INAF), Université Laval, Quebec City, QC G1V 0A6, Canada; rodrigue.daassi.1@ulaval.ca (R.D.); kalvin.durand.1@ulaval.ca (K.D.); 2Chemical Engineering Department, Université Laval, Quebec City, QC G1V 0A6, Canada; denis.rodrigue@gch.ulaval.ca

**Keywords:** electrospray, lignin nanoparticles, response surface methodology, poly(lactic acid) composite, film, food packaging, antioxidant

## Abstract

The development of new processing methods is required in order to meet the continuous demand for thinner films with excellent barrier properties for food packaging and other applications. In this study, rice husk organosolv lignin nanoparticles were prepared using the electrospray method, which were applied to produce polylactic acid (PLA)-based films for food packaging. The effect of the following electrospray parameters has been investigated: lignin concentration (LC) ranging from 5–50 mg/mL, flow rate (FR) from 0.5–1 mL/min, applied voltage from 10–30 kV, and tip-to-collector distance (TCD) from 10–25 cm, on the morphology, size, polydispersity index (PDI), and Zeta potential (ZP) of lignin nanoparticles (LNPs). The response surface methodology with a Box-Behnken design was applied to optimize these parameters, while dynamic light scattering (DLS) and scanning electron microscopy (SEM) analyses were used to characterize the controlled LNPs. The results showed that the LNPs shape and sizes represent a balance between the solvent evaporation, LC, applied voltage, TCD and FR. The application of optimal electrospray conditions resulted in the production of LNPs with a spherical shape and a minimal size of 260 ± 10 nm, a PDI of 0.257 ± 0.02, and a ZP of −35.2 ± 4.1 mV. The optimal conditions were achieved at LC = 49.1 mg/mL and FR = 0.5 mL/h under an applied voltage of 25.4 kV and TCD = 22 cm. Then, the optimized LNPs were used to improve the properties of PLA-based films. Three types of PLA-lignin blend films were casted, namely lignin/PLA, LNPs/PLA and PLA-grafted LNPs. PLA-grafted LNPs exhibited a more uniform dispersion in PLA for lignin contents of up to 10% than other composite samples. Increasing the lignin content from 5% to 10% in PLA-grafted LNPs resulted in a significant increase in elongation at break (up to four times higher than neat PLA). The presence of PLA-grafted lignin led to a substantial reduction in optical transmittance in the UV range, dropping from 58.7 ± 3.0% to 1.10 ± 0.01%, while maintaining excellent transparency to visible light compared to blends containing lignin or LNPs. Although the antioxidant capacity of unmodified lignin is well-known, a substantial increase in antioxidant capacity was observed in LNPs and PLA-grafted LNP films, with values exceeding 10 times and 12 times that of neat PLA, respectively. These results confirm the significant potential of using studied films in food packaging applications.

## 1. Introduction

In recent years, the demand for eco-friendly food packaging with improved properties has increased. Polylactic acid (PLA) has been considered as a promising biopolymer for food packaging due to its renewability, biodegradability, and excellent mechanical properties [1,2]. However, its poor barrier properties against oxygen, water vapor, and UV light, as well as its limited thermal stability and its low elongation at break (mechanical testing), have hindered its practical application [3,4,5]. Therefore, new strategies need to be developed to improve the barrier, thermal, and antioxidant properties of PLA-based food packaging. One approach is the incorporation of natural antioxidants such as lignins which are polyphenolic biopolymers available from pulping processes applied to various vascular plants.

Lignin is the second most abundant natural polymers on Earth, comprising up to 30% of plant biomass [6,7]. Traditionally, lignin has been viewed as a waste by-product of the paper and pulp industry and burned as a low-value fuel. However, in recent years, there has been growing interest in developing lignin-based materials for high-value applications, such as bioplastics, composites, and coatings [8,9,10,11]. Lignins are complex polymers consisting of various phenylpropanoid units and their structures are highly dependent on the plant source, extraction method, and processing conditions [8]. Lignins are plant-based macromolecules with interesting properties such as biodegradability, biocompatibility, antioxidant, and antibacterial characteristics, generally available at low cost [12,13,14]. Lignins antioxidant properties are associated with their free radical scavenging capacity related mainly to free phenolic hydroxyl groups making them promising natural additives to improve the antioxidant properties of food packaging [12]. Moreover, lignins exhibit high pH stability, high thermal stability, and nontoxicity, making them ideal fillers for composite films in food packaging [3]. Additionally, lignins are UV-absorbing polyphenols, which can improve the UV barrier of materials [12]. However, several studies showed that decreasing their particle size to the “nano” level further improved these properties, making them more suitable for nanocomposites production [8,10,11]. Given the promising properties of lignin nanoparticles (LNPs), various strategies have been proposed for their production, including solvent exchange, pH-shift, emulsion templates, and CO_2_ anti-solvent techniques [14,15,16]. However, these methods have several limitations, such as complex and time-consuming procedures, as well as difficult control of the particle size distribution and morphology. In contrast to traditional mechanical or solvent-based procedures for the scalable synthesis of lignin nanoparticles, electrospray techniques have gained momentum in recent years for the fabrication of nanoparticles from polymers with desired size and morphology [14,15,16].

Electrospray is a versatile and straightforward technique that can be used to produce lignin nanoparticles (LNPs) with controlled size and morphology [14]. The electrospray process involves applying a high voltage to a polymer solution, which is then ejected through a capillary [16]. During the process, the solvent evaporates, leaving behind solid nanoparticles. One of the most important factors influencing the electrospray process is the lignin concentration in the polymer solution. Lignin concentration affects several properties of the solution, including viscosity, elasticity, and surface tension, which are controlling the size and morphology of the resulting nanoparticles [14]. Another critical parameter is the flow rate of the polymer solution as it determines the amount of solution ejected through the capillary [15]. Additionally, the high voltage applied during electrospray affects the shape and size of the droplets, which ultimately determine the size and morphology of the resulting nanoparticles [14]. The tip-to-collector distance is also an important factor. This distance determines the time available for solvent evaporation and the formation of solid nanoparticles [14,15,16]. Electrospray techniques have several advantages, such as continuous and single-step processing, minimized solvent usage, reduced particle aggregation, and reduced oxidation of materials during processing [16,17]. Despite these advantages for LNPs synthesis, achieving precise control over the process parameters remains challenging. The challenges are often attributed to the high number of parameters to control and their complex interactions. As such, optimizing the electrospray process and identifying the most effective set of parameters to control LNPs synthesis requires careful investigations.

The blending of lignin with PLA is a promising approach to enhance its properties; however, phase separation can limit its benefits [18,19]. Incorporating LNPs into PLA can also be advantageous due to their high surface-to-volume ratio, which enhances optical and antioxidant properties of composite films [3]. However, the aggregation of LNPs can reduce their potential beneficial effects and negatively impact the diffusion barrier properties of the composite. Effective strategies are required to control the aggregation of LNPs, allowing for the development of the full potential of PLA-LNPs blend to be realized. One such strategy is grafting PLA onto lignin nanoparticles via lactide ring-opening polymerization, which could improve the homogeneity of the interaction between the lignins and the PLA matrix and reduce the occurrence of particle aggregation. To improve the dispersion of LNPs in PLA, the use of lignins nanoparticles grafted with PLA has been investigated in this study.

The electrospray parameters were optimized for the production of lignin nanoparticles from rice husk in order to incorporate them in PLA-based composite. The LNPs produced with the optimized parameters were dispersed into PLA-matrices and the optical, mechanical, thermal, and antioxidant properties of the films were studied and compared to the properties of PLA-grafted nanoparticle blends.

## 2. Materials and Methods

### 2.1. Materials

The ground raw rice husks were obtained from agro-wastes residues as a by-product of a previous project. The RH powder underwent sieving using a mechanical shaker (C. E. Tyler Rotap Sieve 3990, Painesville, OH, USA) to retain only particles within the range of 20 to 60 mesh. The lignin (RHL) was then isolated from rice husk biomass using the organosolv process described by Koumba-Yoya and Stevanovic [20]. Prior to organosolv step, the rice husk underwent acid hydrolysis (a complementary study within the same project) leading to maximum xylose production while limiting its conversion to furfural using conditions optimized by Kasangana et al. [21]. The isolated organosolv lignin (RHL) had a high purity of 96% (2% ash and 2% carbohydrate content), β-O–4′ aryl ether as its major substructure (58.1%), followed by guaiacyl, syringyl, and p-hydroxyphenyl content of 77.4%, 12.9%, and 9.7%, respectively. Poly(lactic acid) (Ingeo^TM^ Biopolymers 4060D) containing 12 mol% D-lactide with a specific weight-average molecular weight (Mw) of 190,000 g/mol was obtained from NatureWorks LLC (Minnetonka, MN, USA). All other required chemicals as dichloromethane (DCM), chloroform (CHCl_3_), 2,2-diphenyl-1-picrylhydrazyl (DPPH) and diazabicyclo [5.4.0]undec-7-ene (DBU) were purchased from Sigma-Aldrich (St. Louis, MO, USA) and were used without further purification. Deionized water was sourced from a Millipore Direct-Q5 ultrapure water system.

### 2.2. Preparation of Lignin Nanoparticles and Experimental Design

Lignin nanoparticles (LNPs) were obtained by electrospray from a lignin solution prepared by dissolving a given amount of organosolv lignin in 95% ethanol. Ethanol was chosen because it has been used in the organosolv process to obtain the RHL as described above. In addition, ethanol has high volatility and relatively low surface tension which is favorable for the formation of a stable Taylor cone in an electrospray experiment [22]. The lignin solution was vortexed and sonicated for 30 min using a CPX 2800 Branson Ultrasonic bath and filtered through a 0.45 µm filter (Fisher Scientific, Cambridge, MA, USA) in order to avoid clogging the syringe needle with the insoluble materials in the electrospray experiment. The initial lignin solution concentration ranged from 5 to 50 mg/mL. Table 1 presents the physical properties of the lignin solutions at various concentrations.

It was observed that the surface tension, viscosity, and electrical conductivity slightly increased with increasing lignin concentration. The electrospray method illustrated in Figure 1 involves loading a conductive lignin solution into a 1 mL syringe HENKE-JECT from Henke Sass Wolf and using a syringe pump (KDS-100; KD Scientific Inc., USA) to inject it through a stainless-steel needle Jensen Global 25 Gauge Blunt (Howard Electronics). The needle is then held at a high positive potential using a high voltage power supply (Spellman, model CZE 1000R). The synthesis of LNPs is influenced by various parameters, including flow rate, high voltage, lignin concentration, and tip-to-collector distance. To achieve controlled lignin nanoparticles, four independent variables were designed and optimized using the response surface method with Box-Behnken design (RSM-BBD). These variables were: (A) lignin concentration (5–50 mg/mL), (B) flow rate (0.5–1 mL/h), (C) operation voltage (10–30 kV), and (D) tip-to-collector distance (10–25 cm). The optimization aimed to minimize the particle size, polydispersity index, and Zeta potential. A total of 27 experimental runs, with two replicates at the center points, were conducted as described in the design matrix from RSM-BBD as presented in Appendix A.

### 2.3. Synthesis of PLA-Grafted Lignin Nanoparticles

PLA-grafted lignin nanoparticles (LNPs-PLA) were synthesized as follows: 300 mg of optimized LNPs, 1.5 g of poly(D,L-lactide) (11.5 mmol), and 50 mg of 1,8-diazabicyclo [5.4.0]undec-7-ene (DBU) (0.33 mmol) were added to a 50 mL Schlenk flask. Next, to which 25 mL of anhydrous dichloromethane (DCM) was added and the reaction mixture was stirred under N_2_ at room temperature for 3 h. The reaction was then quenched with 0.5 mL of acetic acid and the product was precipitated by adding 160 mL of methanol. The final product was obtained by centrifugation (8000 rpm, 20 min, three times) and dried overnight in a vacuum oven at 60 °C [3].

### 2.4. Preparation of PLA Lignin, Lignin Nanoparticles (LNPs) and PLA Grafted LNPs Films

PLA composite films consisting of lignin, lignin nanoparticles (LNPs), and PLA grafted LNPs were prepared by solvent casting. The films contained 1%, 5%, and 10% (wt.) of nanoparticles. After drying overnight at 40 °C in a vacuum oven, PLA was dissolved in chloroform at 10% (wt.) while stirring at room temperature for 24 h. Concurrently, samples containing lignin, LNPs, or PLA-grafted LNPs were stirred in chloroform (10 mg/mL) for 24 h. The lignin dispersion and PLA solutions were then mixed together and stirred for 12 h. The mixture was cast onto a glass petri dish and the lignin content in the composite films was adjusted by varying the volumes of lignin dispersion and PLA solution. The film was left to dry for 48 h at room temperature and then for another 48 h in a vacuum oven at 45 °C.

### 2.5. Characterization of Lignin Nanoparticles and PLA Blend Composite Films

#### 2.5.1. Particle Size and Surface Morphology

Dynamic light scattering (DLS) was used to determine the particle size and polydispersity indices of LNPs. The measurements were performed on a Malvern Zetasizer Nano ZS instrument (Malvern Instruments Ltd., Malvern, UK) using the same solution on which the electrospray experiment was performed (Figure 1b). The Zeta potential of the LNPs was measured using laser Doppler electrophoresis with the same Zetasizer device. Scanning electron microscopy (SEM) with a JEOL-JSM-6360 (JEOL Inc., Pleasanton, CA, USA) and transmission electron microscopy (TEM) with a JEOL-JEM-1230 (JEOL-USA Inc., USA) were used to analyze the morphology, size, and size distribution of LNPs. The SEM images were processed by ImageJ software (Version 1.53e) to obtain the particle size distribution. Samples for SEM and TEM were prepared by collecting particles for 2 min each on a smooth-surfaced cylindrical metal and a carbon-coated copper grid, respectively.

#### 2.5.2. Fourier Transform Infrared (FTIR) Analysis and UV-Visible Spectroscopy of LNP and PLA Blend Films

FTIR spectra of the lignin nanoparticles and PLA blend films were obtained using a PerkinElmer Spectrum 400 spectrometer. The spectra were recorded at a resolution of 4 cm^−1^ from 4000–650 cm^−1^ with 64 scans per sample.

The transparency and UV barrier properties of PLA blend films were determined by measuring the light transmission spectra in the wavelength range of 200–700 nm. The lignin content of PLA-grafted LNPs solutions (wt.%) was determined by recording the UV absorbance at 280 nm using a calibration curve (Appendix A) which was generated by recording the UV absorbance at 280 nm of LNPs solutions in ethanol (concentrations 0–50 μg/mL) using a Varian Cary 50 UV-Vis spectrophotometer (Agilent, Santa Clara, CA, USA).

#### 2.5.3. Thermal Analysis of Lignin and PLA Blend Films

Thermogravimetric analysis (TGA) was performed using a TGA/SDTA851 (Mettler Toledo, USA) under a nitrogen atmosphere (50 mL/min) by increasing the temperature from 30 to 850 °C at a heating rate of 10 °C/min. The derivative of the TGA curves (DTG) was also analyzed.

Differential scanning calorimetry (DSC) measurements were performed using a Mettler Toledo DSC822e under a nitrogen atmosphere (50 mL/min) by increasing the temperature from 30 to 180 °C with a heating rate of 10 °C/min. For each sample, an initial heating scan was performed from 25 to 220 °C at a rate of 10 °C/min to eliminate any heat history. This first cycle was followed by a rapid cooling to 25 °C in 5 min before a second heating cycle from 25 to 220 °C at 10 °C/min. Determination of the glass transition temperature (Tg) was carried out by analyzing the inflection point, corresponding to a slope change during the phase transition as observed in the second heating step.

#### 2.5.4. Mechanical Properties

The tensile properties of the films were analyzed using a dynamic mechanical analyzer (DMA) RSA III from TA Instruments (New Castle, DE, USA). The specimens had a thickness and width of 0.12 and 6.84 mm, respectively. At a constant speed of 1.2 mm/min, at least four measurements were conducted for each sample. The average values of tensile strength, Young’s modulus, and elongation at break were obtained from the respective stress and strain curves.

#### 2.5.5. Antioxidant Activity

Antioxidant capacities of the films was evaluated using the 2,2-diphenyl-1-picrylhydrazyl (DPPH) colorimetric assay with a slight modification as reported by Blois [23]. To start, 100 mg of each film was cut into small pieces and submerged in 4 mL of 25 mg/L DPPH solution in methanol. The samples were shaken for 4 h at room temperature in the dark before measuring their absorbance at 517 nm in triplicate. The DPPH radical-scavenging capacities of the studied films were calculated as the antioxidant capacity (%) as:(1)$$\mathrm{Antioxidant}\mathrm{capacity}\left(\%\right)=\frac{A_{0}-A_{t}}{A_{0}}\times 100$$
where A_0_ and A_t_ are the absorbances at 517 nm of the pure DPPH solution and DPPH solution with dissolved films after 4 h of incubation, respectively.

### 2.6. Statistical Analysis

The effect of each variable on the LNPs preparation via electrospray was investigated using response-surface methodology with Box-Behnken designs (RSM-BBD). A total of 27 experimental runs with two replicates at the centre points were carried out and the design matrix from RSM-BBD is reported in Appendix A. The polynomial Equation for the response variables is:(2)$$Y=\beta_{0}+\sum \beta_{i}X_{i}+\sum \beta_{\mathrm{ij}}X_{i}X_{j}+\sum \beta_{\mathrm{ii}}{X_{i}}^{2}$$
where Y is the response value; β_0_ is the model intercept coefficient; β_i_ is the linear effect; β_ij_ is the interaction effect; β_ii_ is the quadratic effect, while X_i_ and X_j_ are the independent variables.

RSM—BBD experiment design and statistical analyses were performed using the Minitab 22.0 (Minitab, State College, PA, USA) software. The latter was also used to determine the model equation, get the 3-D plot of the response, and predict the optimum values for the response variables. In addition, an analysis of variance (ANOVA) was performed on the data. The results are presented as the mean ± its standard deviation. The least significant difference (LSD) test was conducted to determine significant differences between treatments.

## 3. Results and Discussion

### 3.1. Effect of Lignin Concentration, Flow Rate, Voltage, and Tip-to-Collector Distance on the Synthesis of Rice Husk Lignin Nanoparticles by Electrospray

Several parameters are known to control the stable electrospray production of lignin nanoparticles with reproducible size and morphology. SEM micrographs of LNPs synthesized by electrospray with different levels of lignin concentration (LC), flow rate (FR), voltage, and tip-to-collector distance (TCD) are shown in Figure 2. It can be seen that the LNPs have irregular (Figure 2A), non-spherical (Figure 2B), and spherical shapes (Figure 2C), when increasing the lignin concentration from 5 mg/mL to 27.5 mg/mL, and 50 mg/mL respectively, under variations of other parameter levels. The particles became more agglomerated with a more uniform spherical shapes when LC = 50 mg/mL, FR = 0.5 mg/h, applied voltage = 20 kV and TCD = 10 cm (Figure 2D). The formation of a macromolecular shell on the droplet’s surface, followed by the diffusion of the remaining solvent are making the LNPs more spherical. The morphology of the LNPs is influenced by a competition between the diffusion of lignin in solution and solvent evaporation during electrospray [14]. The accumulation of lignin at the droplet’s surface is due to the short time allowed for lignin to diffuse towards the droplet centre during evaporation [14,24]. This phenomenon is more pronounced when the lignin diffusion rate is slower than the solvent evaporation rate. Additionally, increasing the lignin solution concentration resulted in larger LNPs from 104 ± 30 nm to 152 ± 45 nm, and up to 207 ± 21 nm by decreasing the applied voltage and TCD (Figure 2E). This size increase can be attributed to higher viscosity (Table 1) of the lignin solutions, as reported by other studies [14,25].

### 3.2. Experimental Optimization of LNPs Using a Box-Behnken Design

#### 3.2.1. Effect of Interactive Process Parameters

To optimize the synthesis of LNPs via electrospray, a preliminary test was conducted to select the working range of significant factors. The selected factors were lignin concentration (LC) ranging from 5–50 mg/mL, flow rate (FR) of 0.5–1 mL/h, applied voltage of 10–30 kV, and distance to-tip-collector (TCD) of 10–25 cm. A Box-Behnken design model was used to perform a total of 27 randomized experiments with two replicates to optimize the electrospray parameters to search for minimum LNP size, minimum polydispersity index (PDI), and minimum Zeta potential (ZP). The experimental and predicted results are presented in Appendix A. Multiple regression analysis was conducted based on the response values in Appendix A to obtain the model Equations of LNPs size, PDI, and ZP named (3), (4) and (5) respectively:Z-Ave (d. nm) = 1427 − 18.02 X_1_ − 1584 X_2_ − 20.8 X_3_ − 44.3 X_4_ + 0.3811 X_1×1_ + 1226 X_2_X_2_ + 1.525 X_3_X_3_ + 2.606 X_4_X_4_ + 12.76 X_1_X_2_ − 0.458 X_1_X_3_ − 0.247 X_1_X_4_ − 2.7 X_2_X_3_ − 4.9 X_2_X_4_ − 1.467 X_3_X_4_
(3)
PDI = 2.347 − 0.0061 X_1_ − 2.64 X_2_ − 0.0281 X_3_ − 0.0829 X_4_ + 0.000067 X_1_X_1_ + 0.848 X_2_X_2_ + 0.000639 X_3_X_3_ + 0.001289 X_4_X_4_ + 0.01337 X_1_X_2_ − 0.000117 X_1_X_3_ − 0.000349 X_1_X_4_ + 0.0082 X_2_X_3_ + 0.0599 X_2_X_4_ + 0.000130 X_3_X_4_
(4)
ZP (mV) = −66.1 + 0.903 X_1_ − 5.0 X_2_ + 1.35 X_3_ + 1.03 X_4_ + 0.00071 X_1_X_1_ + 13.8 X_2_X_2_ + 0.0016 X_3_X_3_ + 0.0064 X_4_X_4_ + 0.153 X_1_ ∗ X_2_ − 0.02462 X_1_X_3_ − 0.0294 X_1_X_4_ − 0.795 X_2_X_3_ − 0.36 X_2_X_4_ − 0.0018 X_3_X_4_
(5)
where X_1_, X_2_, X_3_, and X_4_ are the independent variables; i.e., lignin concentration, flow rate, applied voltage, and tip-to-collect distance respectively.

To validate the regression model, a F-test was conducted and the analysis of variance (ANOVA) for the quadratic model of the response surface is presented in Table 2. The ANOVA results indicate that the model is significant (*p*-value = 0.000) with R^2^ ranging from 68 to 94% (Table 2). These results suggest that the model terms are significant and can be effectively used to predict the size, PDI, and ZP of rice husk lignin nanoparticles synthesized via electrospray.

Figure 3, Figure 4 and Figure 5 present the 3-D surface plots of the interaction factors affecting the size, PDI, and ZP of lignin nanoparticles (LNPs), determined using dynamic light scattering (DLS) analysis. The surface response is a critical parameter for determining the optimum values of the variables. It was observed that all surface plots were convex, indicating that the range of independent variables was well selected. Increasing the lignin concentration (LC) and flow rate (FR) significantly (*p*-value ˂ 0.05) affected the size of LNPs, which increased from 310 ± 20 nm to 760 ± 35 nm at an applied voltage of 20 kV and a distance to-tip-collector (TCD) of 17.5 cm (Figure 3A). A smaller particle size was obtained when a lower FR was used, while larger sizes were obtained at a higher FR due to a competitive mechanism of Coulomb fission. Indeed, Coulomb fission is a process by which an electrically charged droplet or particle breaks up into smaller fragments due to the repulsive Coulomb forces between the like charges. In this mechanism, the electrostatic repulsion between the charges can overcome the surface tension holding the droplet or particle together, causing it to break up into smaller pieces as reported in other studies [16]. The likelihood of Coulomb fission increases with the charge of the droplet or particle, and is influenced by factors such as the solvent properties, temperature, and other experimental conditions. Furthermore, a FR of 1 mL/h resulted in wet-deposited LNPs due to a lack of sufficient time to evaporate all the solvent during the process, which induced the formation of larger particles. Conversely, under the effect of LC and applied voltage, particle size varied somewhat but showed no clear trend at FR = 0.5 mL/h and TCD = 17.5 cm (Figure 3B). The size of LNPs was also significantly influenced by the combined effect of LC and TCD (*p*-value ˂ 0.05) (Figure 3C) and then by the combined effect of applied voltage and TCD (*p*-value ˂ 0.05) (Figure 3F). These results showed that increasing the TCD led to smaller LNPs, which was modulated by the applied voltage. When the TCD increases, the electric field intensity decreases leading to larger LNPs. These results confirm those reported by Morais et al. [16], also reporting that increasing the applied voltage resulted in smaller particle size. However, the average LNPs sizes recorded via DLS analysis were larger than those from SEM analysis due to the formation of a hydration layer around the LNPs in water. This phenomenon was also reported in several other works, indicating the differences in LNPs sizes between DLS and SEM analyses [14,15]. However, at a constant LC of 27.5 mg/mL, the combined effects of FR and voltage (Figure 3D), as well as FR and TCD (Figure 3E) did not significantly influence the size of LNPs. The lack of significant influence on size of LNPs from combined effect of FR, voltage, and TCD could be attributed to the inherent properties of the lignin solution and the dominant mechanisms governing droplet formation and evaporation during electrospraying. The unique physicochemical characteristics of lignin, such as its solubility, viscosity, and molecular weight, may contribute to a relatively narrow size distribution of the resulting nanoparticles, thus overriding the effects of these process parameters [14].

The polydispersity index (PDI) is a crucial parameter in nanoparticle characterization reflecting the size distribution and degree of agglomeration. The PDI of LNPs was evaluated using dynamic light scattering (DLS) analysis to also determine the state of particle agglomeration. A lower PDI indicates a narrower size distribution, while a higher PDI suggests a broader range of particle sizes. Controlling the PDI is crucial for achieving homogeneous nanoparticles with predictable properties and performance. Understanding the influence of various parameters on the PDI during the electrospray process for LNPs is essential for optimizing the synthesis conditions. The ANOVA results show that only the combined effect of LC and TCD (Figure 4C) significantly influenced the PDI of LNPs (*p* < 0.05), which overall ranged from 0.209 ± 0.077 to 0.686 ± 0.112 (Figure 4). In particular, increasing the LC and TCD favored a decrease in PDI, indicating a reduction in particle agglomeration. Notably, low LC resulted in reduced interactions between lignin molecules, resulting in negatively charged LNPs that were less prone to agglomeration. These findings are in line with previous studies reporting that particle size and agglomeration can be influenced by different factors and their interactions [24]. It is expected that altering the LC and FR would affect the PDI of the resulting nanoparticles [15]. However, our study showed that changes in LC and FR (Figure 4A) do not significantly influence the PDI of LNPs. This observation suggests that the electrospray process for LNPs is robust and resilient to variations in these parameters when it comes to PDI control. Investigations focusing on the combined effects of LC and voltage (Figure 4B), FR and voltage (Figure 4D), voltage and TCD (Figure 4E), voltage and TCD (Figure 4F) demonstrated no substantial influence on the PDI of LNPs. This finding suggests that other factors, such as the inherent properties of lignin and the dominant mechanisms governing droplet breakup and evaporation, are more significant in determining the PDI.

Zeta potential is a measure of the electrostatic potential at the nanoparticle surface and provides insights into their stability and dispersion characteristics. The Zeta potential was highly influenced (*p* value ˂ 0.05) by the combined effect of LC and applied voltage (Figure 5B) of LC and TCD (Figure 5C) and of FR and applied voltage (Table 2, Figure 5D). All the samples exhibited a negative charge, and the Zeta potential ranged overall from −20.9 ± 4.8 mV to −39.9 ± 3.25 mV (Figure 5). These values are consistent with those reported by previous studies [26,27,28]. Our results showed that increasing LC, applied voltage and TCD with lower FR resulted in lower negative ZP values suggesting LNPs stability (Figure 5). As the ZP becomes more negative, intermolecular electrostatic repulsions increase and the LNPs becomes more stable. The highly negative ZP values indicate that the particles have high dispersibility due to repulsion between LNPs minimizing aggregation. However, our result showed that the combined effects of LC and FR, FR and voltage, and voltage and TCD, had no significant influence on ZP of LNPs during the electrospray process (Figure 5A,E,F). This observation suggests that other factors, including the intrinsic properties of lignin solution, the dynamics of droplet breakup and evaporation, may exert a more dominant role in shaping on the ZP as reported by other studies [29,30].

#### 3.2.2. Model Validation and Confirmation

The optimum conditions were determined using the Minitab software. The synthesis of rice husk LNPs with minimum size, minimum PDI, and minimum ZP was achieved under the following optimum conditions: lignin concentration of 49.1 mg/mL, flow rate of 0.50 mL/h, voltage of 25.4 kV, and tip-to-collector distance 22.0 cm. Under these conditions, the predicted (experimental) values of LNPs size, PDI, and ZP are 284.2 nm (260.3 ± 10.1 nm), 0.241 (0.257 ± 0.02), and −31.7 mV (−35.2 ± 4.1 mV), respectively (Table 3). It can be observed that the experimental values are in good agreement with the predicted values for all parameters indicating that the regression model was satisfactory and accurate. The composite desirability was 0.86 which represents a highly desirable, or ideal response value of these optimum conditions. The ZP of the optimized LNPs (35.2 ± 4.1 mV) indicates the stability of the nanoparticles with less aggregation and better dispersion. The SEM and TEM micrographs of the optimized LNPs reveal that the particle shape was spherical with a smooth surface (Figure 6A,C). The particle size distribution from the SEM micrograph is Gaussian with an average size of 104 ± 24 nm (Figure 6B). The uniform size, low PDI, regular spherical geometry, and high stability of the optimized LNPs suggest that they are promising candidates to produce PLA nanocomposite films.

### 3.3. Characterization of PLA Blend Composite Films

#### 3.3.1. FTIR and Optical Properties

Figure 7 presents the FTIR spectra of the initial materials (lignin, LNPs, and PLA) and of the produced composite films. These FTIR spectra exhibit distinctive peaks in the range of 4000–600 cm^−1^. First, lignin and LNPs have nearly identical spectra and the high similarity between these two samples indicate that only minor chemical changes occurred during the LNPs synthesis via electrospray. The peak around 3415 cm^−1^ can be attributed to the aliphatic and aromatic –OH groups. Transmittance peaks in the region between 3000 and 2842 cm^−1^ were assigned to the stretching of the C–H bonds in the –OCH_3_ groups of lignin. In addition, the transmittance peaks at 1602 cm^−1^ and 1514 cm^−1^ are attributed to the typical aromatic skeletal vibrations [31]. For the PLA film, the strong peak at 1750 cm^−1^ is assigned to the –C=O stretch in ester groups. The –OH groups of lignin have a strong propensity to establish hydrogen bonds with the carbonyl group of PLA molecules by elevating the stretching vibration frequency of the carbonyl group to 1750 cm^−1^ [2,31]. The peaks at 1080 cm^−1^ and 2996 cm^−1^ are attributed to the stretching of the C–O and –CH_3_ bonds, respectively [31]. For the series of PLA composites (lignin/PLA, LNPs/PLA, and LNPs-PLA/PLA), a slight shift to lower wavenumbers was observed, indicating interfacial interactions between the lignin particles and PLA. Moreover, the absence of peaks related to lignin nanoparticles (LNPs) in the FTIR spectra of samples containing both PLA and lignin can be also attributed to the interfacial interactions (low-energy intermolecular interactions). The interfacial interactions between lignin particles and PLA can result in shifts or modifications in the characteristic peaks for both components. These interactions can lead to the formation of new bonds or changes in the vibrational modes of the functional groups involved. As a result, the peaks associated with LNPs may undergo shifts or broadening, making them less distinguishable in the FTIR spectrum of the composite film. These filler-polymer interactions were substantially more important for the LNP-based samples. However, the variations in form and position of the peaks were more prominent for the PLA-grafted LNPs film samples, which could be attributed to the formation of new bonds between the –OH groups of lignin-based samples and the functional groups of PLA.

#### 3.3.2. Optical Properties

The results of optical properties measurements are presented in Figure 8A,B. The transmittance at 660 nm (T660) is an indicator of the film’s transparency in the visible range, while the transmittance at 280 nm (T280) reflects the film’s ultraviolet barrier properties. These two properties are crucial for food packaging applications since the packaging material must provide adequate protection against UV light which is known to degrade proteins, lipids, and vitamins, thus reducing the shelf life of food products [3]. At the same time, the packaging material should be transparent to provide visibility of the product inside. PLA composite films are typically used for food packaging and must provide appropriate optical properties. Lignin surface functional groups, such as va<arious chromophores (ketones, and phenols) can enhance the UV absorbance of lignin-containing composites [31].

Our results show that neat PLA films have high transparency (Figure 8C) as reflected in the high T660 value of 83.1 ± 4.0%. However, as the lignin concentration increased, the T660 value significantly decreased to 20.1 ± 2.0%, indicating reduced transparency. Interestingly, the PLA-grafted LNPs blend films exhibited higher transparency than the LNPs/PLA and lignin/PLA blend films, indicating that the PLA-grafted LNPs were more uniformly distributed within the PLA matrix. On the other hand, the T280 value significantly decreased from 58.7 ± 3.0% to 1.1 ± 0.01% with increasing lignin concentration, indicating that all composite films, especially those containing 10% lignin, had excellent UV barrier properties. This excellent property is attributed to the absorption of conjugated carbonyl and aromatic groups in lignin [2,3]. The optical micrographs of the composites shown in Figure 8C also confirm the transparency results obtained from the transmittance measurements. Even at relatively high lignin concentrations, the PLA blends with PLA-grafted LNPs appear to be more transparent than the PLA blends with LNPs or lignin. This suggests that the PLA-grafted LNPs were better dispersed within the PLA matrix than the LNPs or lignin (Appendix A), which resulted in less phase separation and aggregation. It is worth noting that the organosolv lignin used in this study showed excellent UV protection compared to other lignins examined in similar tests such as alkali lignin and sodium lignosulfonate [32]. These findings make the LNPs-PLA blend films promising candidates for UV-protective food packaging materials. These composite films offer excellent UV barrier properties and can protect the food from harmful UV radiation, while maintaining sufficient transparency to allow the consumers to see the product inside. The uniform distribution of the filler particles within the matrix is crucial to achieve these desirable optical properties, and our study suggests that PLA-grafted LNPs are the most effective.

#### 3.3.3. Thermal Properties of Lignins and PLA-Lignin Composite Blends

The thermal behavior of lignin and PLA films containing 10% lignin was also investigated. Figure 9 presents the typical TGA, DTG, and DSC thermograms of the studied samples. The TGA thermograms show the weight loss pattern of the samples during thermal decomposition, while the DTG thermograms show the maximum decomposition temperature at each step of thermal decomposition. The obtained rice husk results indicate that the initial weight loss peaks are around 105 °C due to water evaporation [6]. The absence of mass loss between 150–250 °C for lignin and LNPs confirms that the rice husk organosolv lignin used for the films was of high purity (very low contents of ashes and of residual carbohydrates). Furthermore, the main degradation step in the range of 305–385 °C is observed in the thermograms of all studied samples, with a maximum degradation rate around 350–360 °C, which is attributed to the decomposition of polymers (lignin and PLA) [2]. Interestingly, higher degradation temperatures were determined for the lignin/PLA, LNPs/PLA, and LNPs-PLA/PLA blend films than for the neat PLA films. This suggests that grafting of PLA to LNPs and the lignin content of the blends increase the film thermal stability. This increased thermal stability of lignin-containing blends confirms the higher heat stability of lignins compared to the PLA matrix.

DSC analysis was used to determine the glass transition (Tg) of lignin and PLA composite films. Tg is a key parameter to consider when using unmodified lignin and LNPs as precursors for composite materials since it provides a direct indication of the mobility of the macromolecular structures [33]. The Tg range reported for the industrial lignins is between 90 and 180 °C [33]. The DSC results (Appendix A) show the similar values determined for lignin and LNPs for Tg (about 164 °C) which is close to the values reported from other studies [5,6,31]. However, the Tg of neat PLA (64.2 °C) was found to be slightly higher than that of PLA composite films (from 62.7 °C to 63.2 °C), which could be attributed to the plasticizing effect of a low molecular weight fraction of lignin [5]. Therefore, our results suggest that the addition of lignin or LNPs has a notable impact on the Tg of PLA composite films. This could have substantial implications for their potential applications in food packaging materials, as demonstrated by their optical properties (Figure 8). The addition of lignin and LNPs introduces new molecular interactions within the PLA matrix, leading to changes in its overall Tg. Lignin, being a natural polyphenolic compound, possesses functional groups that can interact with PLA chains, affecting their mobility and subsequently altering the Tg of the composite films. The presence of LNPs further enhances these interactions due to their higher surface area and potential for increased contact with the PLA matrix.

#### 3.3.4. Mechanical Properties

The mechanical properties of films are crucial to determine their potential as packaging materials to preserve their structure under mechanical stresses in various applications [31]. These properties were studied via tensile testing. Figure 10 presents the tensile strength, Young’s modulus, and elongation at break for lignin-PLA blends, obtained from tensile stress–strain curves as illustrated in Appendix A. Neat PLA exhibits a fragile behavior with a low elongation at break (about 1.7%) and a high Young’s modulus (4.7 GPa). These results corroborate those of previous work published on PLA films prepared by solvent casting [3]. The tensile strength and Young’s modulus of PLA blends slightly decreased compared to neat PLA, but did not significantly change at different lignin (lignin, LNPs, PLA-grafted LNPs) contents. The elongation at break of the PLA blend film did not change when 1% of lignin, LNPs and PLA grafted LNPs was incorporated but increased significantly when the lignin content was increased to 5% for the lignin PLA blend, LNP and LNP-PLA. However, increasing the lignin content up to 10% resulted in lower elongation at break for lignin/PLA and LNPs/PLA. This could be due to phase separation and the weak cohesion/adhesion between PLA and lignin in these films. On the other hand, by increasing the lignin content from 5 to 10%, LNPs-PLA/PLA revealed an additional increase in elongation at break up to 4 times greater than that determined for neat PLA. This could be assigned to better interfacial interactions due to the grafted PLA molecules acting as compatibilizers.

#### 3.3.5. Antioxidant Capacity of Lignin-PLA Blends

The antioxidant capacity of PLA blend films was determined using the DPPH assay. DPPH is a free radical having very high absorbance at 517 nm and becoming colorless upon radical scavenging reactions with the studied substrates [3]. The decrease in absorbance intensity at 517 nm allows for the calculation of DPPH conversion, which is considered as a measure of the film antioxidant activity (radical scavenging capacity). Figure 11 and Appendix A present the antioxidant activity and UV-vis spectra of DPPH solutions exposed to the neat PLA and PLA/lignin blend samples studied, respectively. The results show that the neat PLA film exhibits low radical scavenging capacity, while the addition of 1% lignin to the PLA blend highly increased this capacity. In addition, the antioxidant capacity of the studied blends increased with increasing lignin content. Furthermore, at low lignin content (1%), the PLA-grafted LNPs film performed worse than the PLA blends with lignin or LNPs. This could be related to the good dispersion of lignin and LNPs in the PLA matrix. Thus, the effect of higher number of free OH groups introduced through higher lignin and LNPs concentrations dominates the effect of the dispersion of PLA-grafted LNPs, since some of the free phenolic OH groups which served as initiation sites for the grafting with the lactide during the formation of PLA-grafted LNPs were used and are no longer available for radical scavenging activity. However, at 5–10% lignin content, the PLA-grafted LNPs films showed higher antioxidant activity than the PLA-lignin and the LNPs blend films. This could be attributed to the effects of LNPs aggregation and lignin phase separation becoming more important, while PLA grafted LNPs remain well dispersed. Therefore, the antioxidant activity of PLA-grafted LNPs is higher than that of other composites despite the reduced number of free OH groups available in these grafted PLA samples. In addition, at 5–10% lignin content, the antioxidant capacity of the LNPs film is higher than that of lignin film samples. Several studies reported that antioxidant capacity increased when lignin was transformed into nanoparticles [1,3]. Although unmodified lignin exhibits favorable antioxidant capacity, the significant increase, in antioxidant activity observed in LNPs and PLA-grafted LNP films, with values more than 10 times and 12 times that of PLA, respectively, highlights their high potential for food packaging applications.

## 4. Conclusions

This study demonstrated that the controlled production of rice husk (RH) lignin nanoparticles (LNP) via electrospray can be optimized using the response surface methodology (RSM) combined with a Box-Behnken design. Several parameters were investigated, including lignin concentration, solution flow rate, applied voltage, and tip-to-collector distance. Each parameter (alone or in combination) was found to have a significant effect on the shape, size, polydispersity index (PDI), and Zeta potential (ZP) of the produced LNPs. The optimized nanoparticles were produced with a spherical shape and uniform distribution using the following conditions: 49.1 mg/mL of lignin concentration in ethanol, 0.5 mL/h of flow rate, 25.4 kV of applied voltage, and 22.0 cm of tip-to-collector distance. The uniform size distribution, low PDI, high negative ZP, regular spherical structure, and high stability of the optimized LNPs highlight their potential as promising candidates for the production of PLA nanocomposite films. The optimum nanoparticles were then used to prepare PLA-lignin blends. Lignin and lignin nanoparticles were shown to increase the thermal stability of PLA composite films, as lignin was more heat stable than PLA. Increasing the lignin content from 5 to 10% in PLA grafted LNPs also increased the elongation at break by more than 4 times, while efficiently decreasing the optical transmittance in the UV range from 58.7 ± 3.0% to 1.10 ± 0.01% with good transparency to visible light. Finally, incorporation of LNPs grafted onto PLA led to enhanced visible light transmission and antioxidant properties by more than 12 times compared to the neat matrix. These results clearly showed the potential of composite films based on PLA-grafted LNPs for food packaging applications.

## Figures and Tables

**Figure 1 polymers-15-02973-f001:**
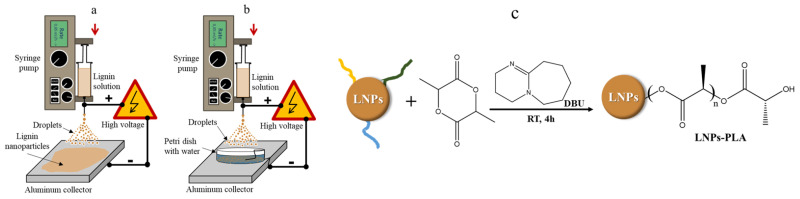
Schematic diagram of lignin nanoparticle synthesis by electrospray and grafting of PLA. (**a**): electrospray with an aluminium plate as collector, (**b**): electrospray with a petri dish containing deionized water under electric field, (**c**): grafting of PLA via lactide ring-opening polymerization.

**Figure 2 polymers-15-02973-f002:**
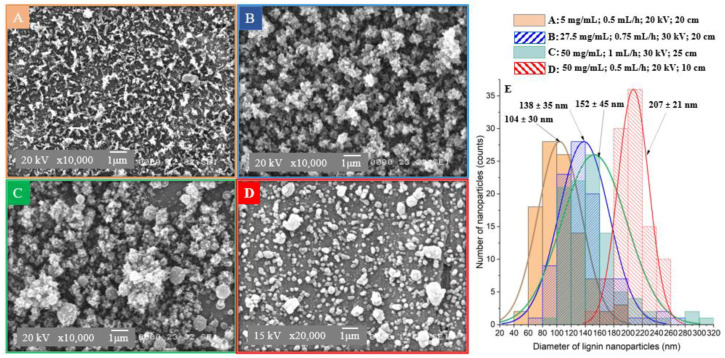
Effect of lignin concentration and flow rate on the morphology and size of lignin nanoparticles (LNPs). SEM micrograph of LNPS synthetized at: (**A**): 5 mg/mL, 0.5 mL/h, 20 kV, 20 cm; (**B**): 27.5 mg/mL, 0.75 mL/h, 30 kV, 20 cm; (**C**): 50 mg/mL, 1 mL/h, 30 kV, 10 cm; (**D**): 50 mg/mL, 0.5 mL/h, 20 kV; 10 cm. (**E**): LNPs particle size distribution obtained from SEM micrographs (**A**–**D**).

**Figure 3 polymers-15-02973-f003:**
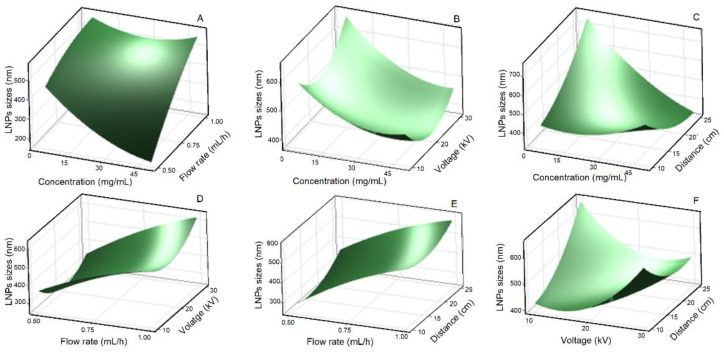
Response surface plot of LNPs size for the effect of lignin concentration and flow rate (**A**), lignin concentration and applied voltage (**B**), lignin concentration and tip-to-collector distance (**C**), flow rate and applied voltage (**D**), flow rate and tip-to-collector distance (**E**), and applied voltage and tip-to-collector distance (**F**).

**Figure 4 polymers-15-02973-f004:**
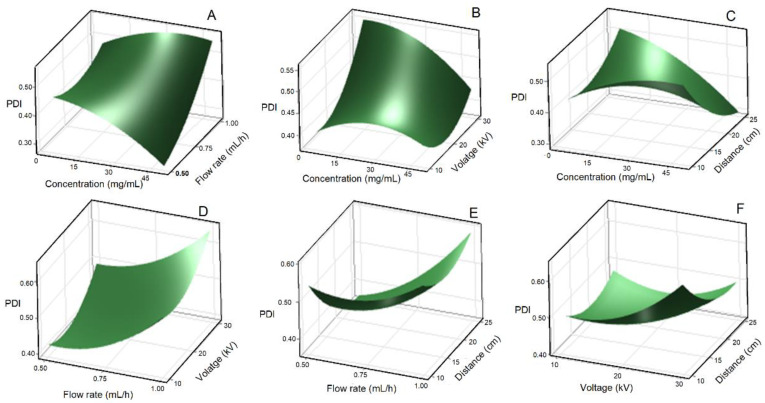
Response surface plot of polydispersity index for the effect of lignin concentration and flow rate (**A**), lignin concentration and applied voltage (**B**), lignin concentration and tip-to-collector distance (**C**), flow rate and applied voltage (**D**), flow rate and tip-to-collector distance (**E**), and applied voltage and tip-to-collector distance (**F**).

**Figure 5 polymers-15-02973-f005:**
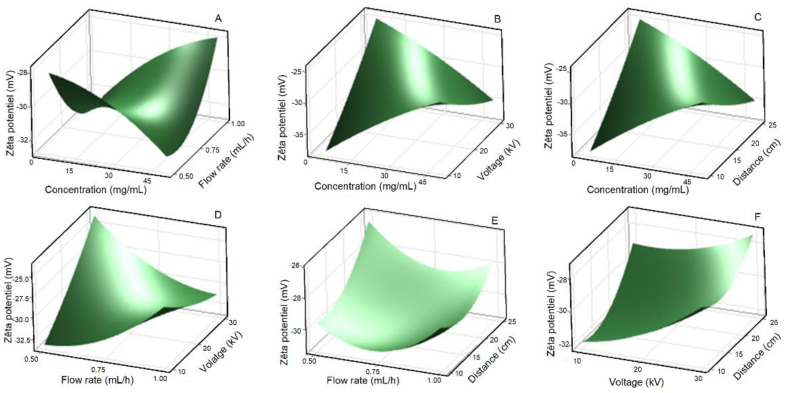
Response surface plot of Zeta potential for the effect of lignin concentration and flow rate (**A**), lignin concentration and applied voltage (**B**), lignin concentration and tip-to-collector distance (**C**), flow rate and applied voltage (**D**), flow rate and tip-to-collector distance (**E**), and applied voltage and tip-to-collector distance (**F**).

**Figure 6 polymers-15-02973-f006:**
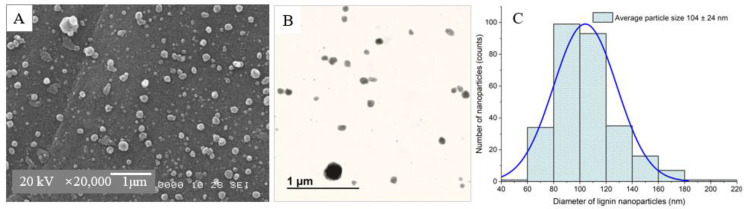
SEM micrograph (**A**), TEM micrograph (**B**), and particle size distribution from SEM micrograph (**C**) of the optimized lignin nanoparticles from rice husk.

**Figure 7 polymers-15-02973-f007:**
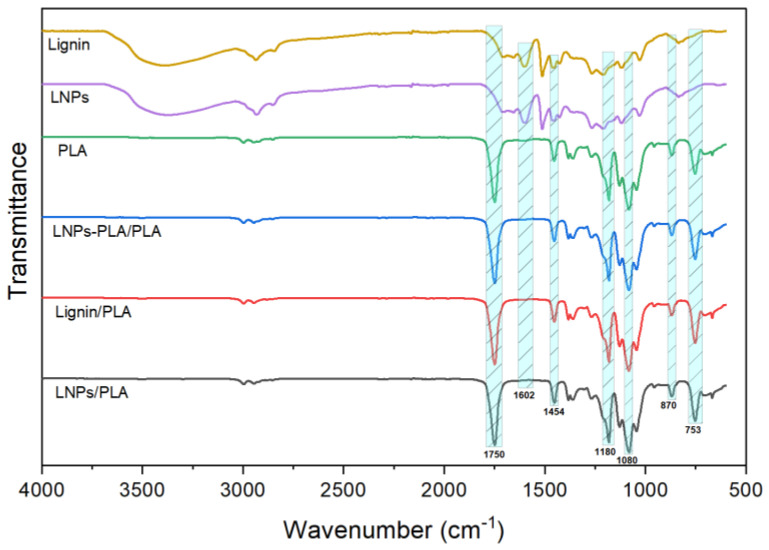
Fourier-transform infrared spectra of polylactic acid (PLA)-based composite films with 10% of lignin.

**Figure 8 polymers-15-02973-f008:**
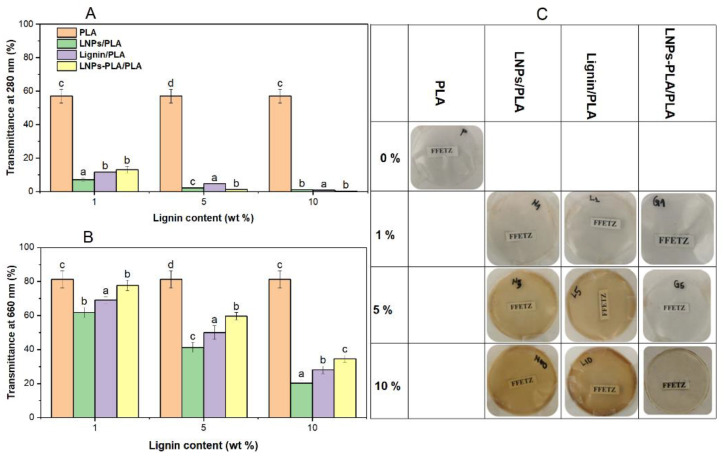
Transmittance (%) of neat PLA films and PLA blends containing lignin, lignin nanoparticles (LNPs), and PLA-grafted lignin nanoparticles (LNPs-PLA) with 1, 5, and 10% lignin content, measured at 280 nm (**A**) and 660 nm (**B**); (**C**) images of different films. For the bar plots, the error bars correspond to the average of transmittance results from triplicate. Bar plots with the same letter are not significantly different based on lignin content; LSD test *p* < 5%.

**Figure 9 polymers-15-02973-f009:**
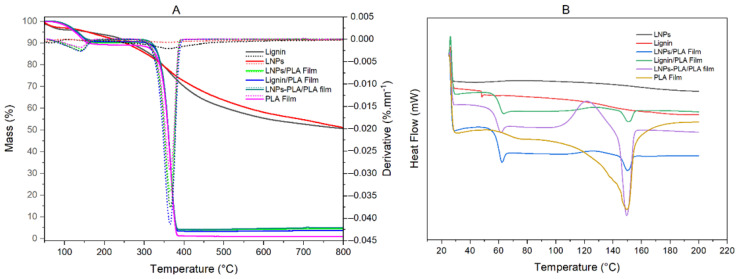
Thermal properties of neat PLA and PLA blend lignin composites, each sample containing 10% lignin: (**A**) TGA and DTG, and (**B**) DSC.

**Figure 10 polymers-15-02973-f010:**
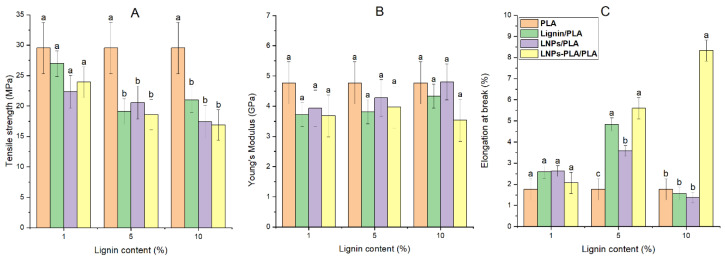
Mechanical properties of neat PLA sample, and film of PLA blend with lignin, lignin nanoparticles, and PLA-grafted LNPs each containing 1, 5, and 10% of lignin. Tensile strength (**A**), Young’s modulus (**B**), and elongation at break (**C**). Bar plot and error bars correspond to average of transmittance results and error type of triplicate. Bar plots with same letter are not significantly different with lignin content; LSD test *p* < 5%.

**Figure 11 polymers-15-02973-f011:**
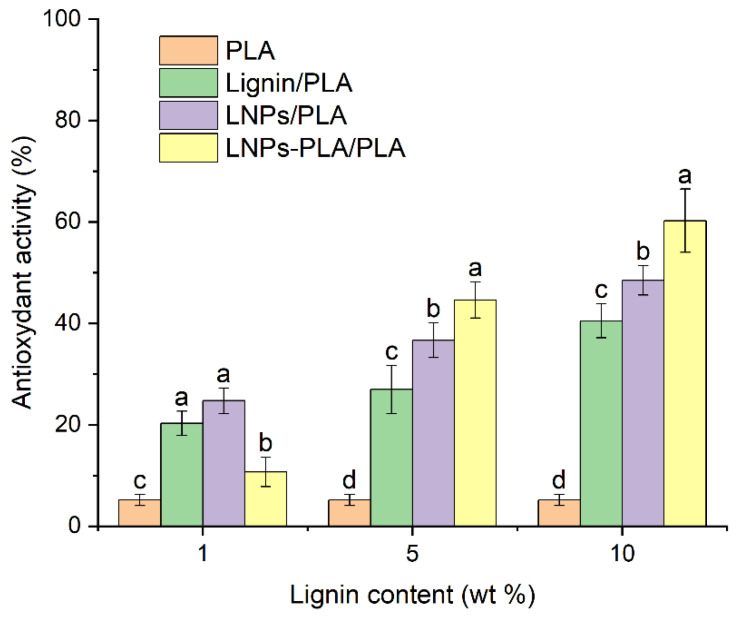
Antioxidant activity of neat PLA films and PLA composites such as PLA with lignin, lignin nanoparticles (LNPs), and PLA-grafted lignin nanoparticles (LNPs-PLA) with 0, 1, 5, and 10 wt.% lignin content, as determined by the DPPH assay. Bar plot and error bars correspond to the average and standard error of measurements in triplicate. Letters compare antioxidant activity of neat PLA films and PLA composites for each lignin content. LSD test *p* < 5%.

**Table 1 polymers-15-02973-t001:** The physical properties of lignin solutions at different concentrations.

Lignin Concentration (mg/mL)	Surface Tension (mN/m) ^a^	Viscosity (cP) ^b^	Electrical Conductivity (μS/cm) ^c^
5	24.68	0.795	17.65
27.5	25.34	1.560	220
49.1 *	24.18 *	1.425 *	106 *
50	25.96	1.957	261

^a^ Surface tension was determined by a Kruss surface and interfacial Tensiometer K10 PST (Kruss, Germany). ^b^ The viscosity was measured using a Cambridge Applied System VL-4100 apparatus. ^c^ The electrical conductivity was measured by a YSI model 35 conductance meter. * measurement of the properties of the lignin solution with the concentration obtained after optimization.

**Table 2 polymers-15-02973-t002:** Analysis of variance of LNPs size, PDI, and Zeta potential for the electrospray experiments.

Source	Sizes Z-Ave (d/nm)				Polydispersity Index (PDI)			Zeta Potential (mV)		
	DF	Mean Square	F Value	*p* Value	Mean Square	F Value	*p* Value	Mean Square	F Value	*p* Value
Model	15	59,572	6.28	0.000	0.027653	3.13	0.000	48.946	4.59	0.000
A: Lignin concentration	1	141,827	14.96	0.001	0.018648	2.11	0.155	0.597	0.06	0.814
B: Flow rate	1	357,204	37.68	0.000	0.116622	13.18	0.001	0.23	0.02	0.884
C: Voltage	1	2800	0.30	0.59	0.03828	4.33	0.044	35.078	3.29	0.077
D: Distance		18,092	1.91	0.175	0.04399	4.97	0.032	26.355	2.47	0.124
AB	1	41,213	4.35	0.044	0.0315	3.56	0.067	35.701	3.35	0.075
AC	1	2615	0.28	0.602	0.00845	0.96	0.335	213.366	20.03	0.000
AD	1	123,529	13.03	0.001	0.048828	5.52	0.024	261.061	24.51	0.000
BC	1	4756	0.50	0.483	0.000171	0.02	0.89	97.301	9.13	0.004
BD		502	0.05	0.819	0.01428	1.61	0.212	2.365	0.22	0.64
CD	1	59,521	6.28	0.017	0.0063	0.71	0.404	1.201	0.11	0.739
A^2^	1	30,078	3.17	0.083	0.012765	1.44	0.237	5.264	0.49	0.486
B^2^	1	7721	0.81	0.372	0.016987	1.92	0.174	29.916	2.81	0.102
C^2^	1	75,245	7.94	0.008	0.024576	2.78	0.104	3.808	0.36	0.553
D^2^	1	27,178	2.87	0.099	0.021123	2.39	0.131	14.719	1.38	0.247
Lack of Fit	34	10,336	4.7	0.07	0.009672	5.24	0.058	11.888	75.28	0.000
R^2^ (X)		94.57			79.54			64.46		
AdjR^2^ (%)		91.72			68.77			50.43		

**Table 3 polymers-15-02973-t003:** Predicted and experimental properties of LNPs synthetized using optimum conditions.

Optimum Condition				Lignin Nanoparticles of Rice Husk					
LC	FR	Applied Voltage	TCD	Size Z-Ave (nm)		PDI		ZP (mV)	
				Predicted	Experimental	Predicted	Experimental	Predicted	Experimental
49.10	0.50	25.4	22.0	284.2	260.3 ± 10.1	0.241	0.257 ± 0.020	−31.7	−35.2 ± 4.1

Experimental data were measured in triplicate.

## Data Availability

Data underlying the results presented in this paper are not publicly available at this time but may be obtained from the authors upon reasonable request.

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
