# Peer review of "Optimization of the Electrospray Process to Produce Lignin Nanoparticles for PLA-Based Food Packaging"

_polymers, 2023, doi:10.3390/polym15132973_

Round 1

Reviewer 1 Report

Reviewed the paper entitled “Optimization of the Electrospray Process to Produce Lignin Nanoparticles for PLA-based Food Packaging”. Paper is very good. In Food Industry particularly for packing it is a hot topic. Content is overall okay, but English and grammar part has to be change. With major revision, I am suggesting some points to be change in the revised version of the paper.

1.      Rewrite the abstract part. Add more numerical data.

2.      Why authors have chosen rice husk as a research material?

3.      Page 2: However, in recent years, there has been growing interest in developing lignin-based materials for high-value applications, such as bio plastics, composites, and coatings. Rewrite the sentences with latest references.

4.      What type of rice husk authors have taken?

5.      What is the need of using acid hydrolysis for rice husk? Explain with latest citation.

6.      Section 2.2: meaning of Error! Reference source not found.

7.      Table 1 is not clear. Reframe the table 1 with all details.

8.      Section 2.3: Why vacuum oven was used and dried overnight in a vacuum oven at 60°C.

9.      Explain Figure 2 properly.

10.  Equation number not cited or aggrieved properly.

11.  Why authors have chosen RSM for optimization? RSM is not recommended most of the cases for optimization as its predicted very is very limited/accurate.

12.  Rewrite the conclusion part.

Okay, extensiveediting of English language required

Reviewer 2 Report

It was a study about the fabrication and evaluation of lignin nanoparticles prepared via the electrospray method and grafting them to PLA polymer for food packaging application. Here are some comments on this study that should be considered before publication:

1-      The scale bar of Figure 2 is not readable.

2-      Please add SEM images of lignin nanoparticles grafted to PLA.

3-      Why is there no difference between the FTIR spectrum of LNPs-PLA and the other two?

4-      The ester group in PLA film resulted from which reaction?

5-      Why has the bond related to the hydroxyl group of lignin disappeared in the spectrum of its complex with PLA? The same for its e C–H bonds.

1-      There are some typo mistakes in the text that should be corrected, some of them are as follows:

·       … ranged from 5 to 50 mg/mL Error! Reference source not found. presents the physical …

·       The electrospray method illustrated in Error! Reference source not found. İnvolves …

·       … experiment was performed (Error! Reference source not found.b).

·       … distance (TCD) are shown in Error! Reference source not found.

·       Additionally, increasing the lignin solution concentration resulted in larger LNPs from 104±30 nm to 152±45 nm (Error! Reference source not found.D).

·       This size increase can be attributed to higher viscosity (Error! Reference source not found.) of the lignin solutions, as reported by other studies

·      

Round 2

Reviewer 1 Report

The Author appropriately incorporates all the suggestions. In this version, the author has given more importance to technical accuracy, so the manuscript may be accepted.

Author Response

The reviewer hat no comment.

Reviewer 2 Report

Thank you so much from the authors for their complete reply, just one comment remains: 

Why we couldn't see peaks related to lignin nanoparticles in the FTIR of samples containing both PLA and lignin? 
